# TransRP: Transformer-based PET/CT feature extraction incorporating clinical data for recurrence-free survival prediction in oropharyngeal cancer

**Baoqiang Ma**[1]                                                                        B.MA@UMCG.NL
**Jiapan Guo**[1 2 3]                                                                     J.GUO@RUG.NL
**Lisanne V. van Dijk**[1 4]                                                         L.V.VAN.DIJK@UMCG.NL
**Peter M.A. van Ooijen**[1 2]                                                 P.M.A.VAN.OOIJEN@UMCG.NL
**Stefan Both**[1]                                                                       S.BOTH@UMCG.NL
**Nanna Maria Sijtsema**[1]                                                    N.M.SIJTSEMA@UMCG.NL

[1]*Departmemt of Radiation Oncology, University Medical Center Groningen, University of Groningen, Groningen, Netherlands*

[2]*Machine Learning Lab, Data Science Center in Health (DASH), University Medical Center Groningen, Groningen, Netherlands*

[3]*Bernoulli Institute for Mathematics, Computer Science and Artificial Intelligence, University of Groningen, Groningen, Netherlands*

[4]*Department of Radiation Oncology, The University of Texas MD Anderson Cancer Centre, Houston, TX, United States*

**Editors:** Accepted for publication at MIDL 2023

## Abstract

The growing number of subtypes and treatment options for oropharyngeal squamous cell carcinoma (OPSCC), a common type of head and neck cancer, highlights the need for personalized therapies. Prognostic outcome prediction models can identify different risk groups for investigation of intensified or de-escalated treatment strategies. Convolution neural networks (CNNs) have been shown to have improved predictive performance compared to traditional clinical and radiomics models. However, CNNs are limited in their ability to learn global features within an entire volume. In this study, we propose a Transformer-based model for predicting recurrence-free survival (RFS) in OPSCC patients, called TransRP. TransRP consists of a CNN encoder to extract rich PET/CT image features, a Transformer encoder to learn global context features, and a fully connected network to incorporate clinical data for RFS prediction. Additionally, we investigated three different methods for combining clinical features into TransRP. The experiments were conducted using the public HECKTOR 2022 challenge dataset, which includes pretreatment PET/CT scans, Gross Tumor Volume masks, clinical data and RFS for OPSCC patients. The dataset was split into a hold-out test set (n = 120) and a training set (n = 362) for five-fold cross-validation. The results show that TransRP achieved the highest test concordance index of 0.698 (an improvement > 2%) in RFS prediction compared to several state-of-the-art clinical and CNN-based methods. In addition, we found that incorporating clinical features with image features obtained from the Transformer encoder performed better than using the Transformer encoder to extract features from both clinical and image features. The code for this study is available at https://github.com/baoqiangmaUMCG/TransRP.

**Keywords:** Transformer, recurrence-free survival prediction, oropharyngeal cancer.

## 1. Introduction

The common treatment of head and neck cancer (HNC) patients is (chemo)radiotherapy with/without surgery. Positron emission tomography (PET) and computed tomography (CT) scans are used in clinical practice to provide functional and structural visualization, enabling oncologists to delineate tumor and organs-at-risk contours for precise radiation dose plans. Oropharyngeal squamous cell carcinoma (OPSCC) is a common type of HNC, and OPSCC patients with human papillomavirus (HPV) have significantly higher 5-year overall survival rates (75%-80%) than tobacco- or alcohol-related patients (45%-50%) (O'Sullivan et al., 2016). Prognostic outcome prediction models are important to optimize treatment strategies for individual patients.

The Tumor-Node-Metastasis (TNM) staging system, including the HPV status in the eighth edition of the American Joint Committee on Cancer (AJCC) guidelines (Amin et al., 2017), is widely used for prognostic outcome prediction in patients with OPSCC (Würdemann et al., 2017). Ang et al. (Ang et al., 2010) published a risk classification system for overall survival based on HPV status, smoking pack years, tumor stage and nodal stage. In addition to clinical models, radiomics features have predictive value for outcomes such as overall survival (OS) and local control (LC) in OPSCC (Haider et al., 2020; Bos et al., 2021). While handcrafted radiomics features are more interpretable to oncologists due to their clear definitions, deep learning techniques can extract more comprehensive and representative image features, potentially obtaining better outcome prediction(Afshar et al., 2019).

Convolution neural networks (CNNs) have been successfully applied to various tasks, including image classification using ResNet (He et al., 2016), DenseNet (Huang et al., 2017) and Efficientnet (Tan and Le, 2019) architectures . A multi-center study on HNC demonstrated that a 2D CNN that inputting the central slice of the primary tumor from a CT image achieved higher area under the curve (AUC) values for 2-year distance metastasis (DM), loco-regional failure (LRF), and OS prediction compared to the radiomics approach (Diamant et al., 2019). With the same dataset, Pang et al. proposed a 3D multi-scale ResNet with a combination of three loss functions and achieved state-of-the-art AUCs of 0.91, 0.78, and 0.70 for 2-year DM, LRF, and OS prediction, respectively (Pang et al., 2022). However, these studies did not address time-to-event outcome prediction and only used pretreatment CT scans, without incorporating clinical predictors.

Several studies have demonstrated that deep learning is effective in time-to-event prediction for OPSCC or HNC. For example, Kim et al. (Kim et al., 2019) proposed to use Deep-Surv (Katzman et al., 2018) model based on clinical parameters to predict OS of OPSCC patients, which outperformed random survival forest (RSF) and Cox proportional hazard (CPH) models. Additionally, other studies utilized imaging data only by CNNs. Cheng et al. (Cheng et al., 2021) proposed a fully automated CNN model for Gross Tumor Volume (GTV) segmentation and OS prediction in OPSCC patients using PET scans. Wang et al. (Wang et al., 2022b) compared the prediction performance of various 3D ResNet-based models trained on different combinations of PET/CT and GTV mask in HNC patients, and found that the PET-only model performed best in DM and OS prediction.

The HECKTOR 2021 challenge (Andrearczyk et al., 2021) provided pretreatment PET/CT scans, clinical data, and GTV of primary tumors (GTVp) for three tasks: (1) GTVp seg-

mentation, (2) progression-free survival (PFS) prediction without GTVp, and (3) PFS prediction using GTVp. The winning team in task 3 used a 3D DenseNet121 (Huang et al., 2017) model to input PET/CT/GTVp and clinical data, and demonstrated that clinical data and GTVp can improve PFS prediction (Wahid et al., 2021). The HECKTOR 2022 challenge (Andrearczyk et al., 2023) included two tasks: (1) GTV of primary tumors and lymph nodes segmentation, and (2) recurrence-free survival (RFS) prediction without GTV. High-ranking teams in task 2 either used clinical and radiomics features based on automatic GTV segmentation by CNNs (Rebaud et al.; Wang et al., 2022a) or a multi-task CNN-based model for GTV segmentation and RFS prediction (Meng et al., 2022). In particular, Meng et al. used residual and dense blocks in their RFS prediction model (Meng et al., 2022). These studies suggest that CNN-based models with advanced blocks such as ResNet or DenseNet and utilizing clinical and CT/PET/GTV images together can achieve good outcome prediction in OPSCC patients.

Although CNNs can extract rich detailed features, they have two major limitations. Firstly, CNNs are limited in modelling explicit long-range relations due to the locality of convolution operations. Secondly, CNNs may be limited in combining multi-modality (clinical and image) information because CNN was designed only for image processing. Vision Transformer (ViT) (Dosovitskiy et al., 2020) could better learn the global context and combine multi-modality features by self-attention mechanism while it is more data-hungary than CNNs. Therefore, pure ViT may not beat CNN models when the training set is not big enough such as medical datasets. Some researchers used CNNs extracting rich image features and a ViT learning global contexts together for the segmentation task (Chen et al., 2021). The combination of CNN and transformer (also used for multi-modality information combination) is also used in cancer outcome prediction. For example, one study used a ViT to learn the interaction between clinical data and image patches and a CNN for PFS prediction in OPSCC patients (Saeed et al., 2022). Another study concatenated clinical data to CNN-extracted CT feature maps and used a ViT to learn global contexts, achieving state-of-the-art survival prediction in nasopharyngeal cancers (Zheng et al., 2022).

Our contributions in this study are summarized as follows: (1) We proposed a novel prediction model named TransRP for RFS in OPSCC patients based on PET/CT/GTV and clinical data. To our best knowledge, we are the first to use Transformer in the RFS prediction of OPSCC. (2) TransRP combines a CNN-encoder for the extraction of rich image features and a Transformer-encoder for learning global contexts. (3) We evaluated the performance of TransRP based on two different CNN-encoders: ResNet18 (He et al., 2016) and DenseNet121 and found that these two TransRP models both showed better prediction than CNN-based models. (4) Transformer was used to combine clinical and image features in two state-of-art studies for HNC and OPSCC outcome prediction (Zheng et al., 2022; Saeed et al., 2022). However, our results found that using a simple fully connected layer to incorporate clinical data can achieve better prediction.

## 2. Methods

The TransRP model in Figure 1(a), includes three main components: a CNN-based encoder, a Transformer-based encoder (ViT), and a fully connected network (FCN). The CNN-based encoder processes PET/CT/GTV volumes to extract image features, which are then passed

to a linear projection and position embedding process for maintaining spatial information. Then, the embedded patches are fed into the transformer encoder which uses self-attention to learn global contexts. The output of the Transformer encoder is then fed to the FCN for the time-to-event prediction of RFS. In the following sections, we will describe the TransRP architecture in further detail and different ways to incorporate clinical data into the TransRP model.

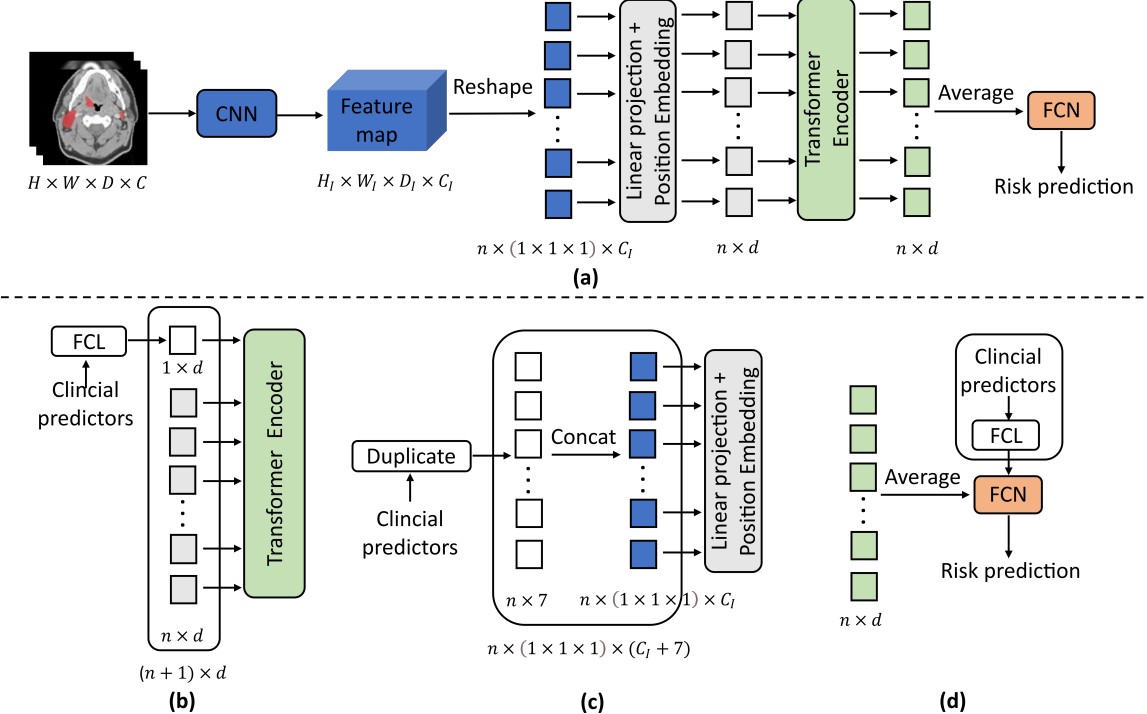

Figure 1: The architecture of the proposed TransRP based on imaging data (a) and three methods of incorporating clinical data into TransRP (b)(c)(d). CNN: convolution neural network. FCN: fully connected network. FCL: fully connected layer

## 2.1. Model architecture

### 2.1.1. CNN-BASED ENCODER

The 3D ResNet18 (He et al., 2016; Hara et al., 2018) and 3D DenseNet121 (Huang et al., 2017) are two options for the CNN encoder in the TransRP model. These models were implemented using the default settings in the MONAI 1.0.0 package with slight adjustments. The detailed architecture of the 3D ResNet18 is shown in Figure 2 of Appendix A, and the 3D DenseNet121 was constructed using the default settings in MONAI 1.0.0, with the exception of setting the kernel and stride to 1 in the average pooling of the third transition layer. Given an input image $x \in R^{H \times W \times D \times C}$ with size $H \times W \times D$ and $C$ channels, the

CNN encoder produces image feature maps $f_{image} \in R^{H_I \times W_I \times D_I \times C_I}$ with size $H_I \times W_I \times D_I$ and $C_I$ channels.

### 2.1.2. PATCH EMBEDDING

Initially, the image feature maps obtained from the CNN-based encoder $f_{image} \in R^{H_I \times W_I \times D_I \times C_I}$ are reshaped to $f_{image} \in R^{n \times (1 \times 1 \times 1) \times C_I}$, where $n = H_I \times W_I \times D_I$. This reshaped feature is then fed to a trainable linear projection, mapping each vectorized $1 \times 1 \times 1$ patch into a $d$-dimensional latent space to obtain patch embeddings $f_{emb} \in R^{n \times d}$. In order to encode the spatial information between the $n$ patches, a learnable position embedding $f_{pos} \in R^{n \times d}$ is added to the patch embeddings to obtain the input for the Transformer encoder: $f_0 = f_{emb} + f_{pos}$.

### 2.1.3. TRANSFORMER ENCODER

The Transformer encoder consists of $L$ layers of multi-head attention (MSA) and multi-layer perceptron (MLP) blocks, as shown in Figure 2 of Appendix A. Therefore, the output of the $i^{th}$ layer of the Transformer encoder is computed as Equation (1) and Equation (2):

$$f_i' = MSA(LN(f_{i-1})) + f_{i-1} \tag{1}$$

$$f_i = MLP(LN(f_i')) + f_i' \tag{2}$$

where $LN()$ is the layer normalization and $f_i$ denotes image representation. In this study, transformer encoder used the same setting with ViT-base (Dosovitskiy et al., 2020).

### 2.1.4. FCN AND LOSS FUNCTION

The output of transformer is averaged over $n$ patches and input to a FCN for RFS prediction. The FCN consists of three layers with $d$, 64 and 1 nodes, respectively.

We adopted the Cox negative logarithm partial likelihood loss proposed in DeepSurv (Katzman et al., 2018), which is shown in Equation (3):

$$L = -\frac{1}{N_{E=1}} \sum_{i:E_i=1} (h_i - log \sum_{j \in R(T_i)} e^{h_j}) + \lambda \|\theta\|_2 \tag{3}$$

where $E$ is the event indicator, $E = 1$ corresponds to a OPSCC patient with event and $E = 0$ indicates a censored patient, $h$ is the predicted risk score by the model, $N_{E=1}$ is the number of all patients with event, $T$ is time-to-event or time-to-censored and $R(T_i)$ is the patients set whose $T$ are not shorter than $i^{th}$ patient who had event. $\theta$ are network parameters and $\lambda$ controls the contribution of $L_2$ regularization.

## 2.2. Incorporating clinical data

In previous studies of Transformer-based models for outcome prediction in patients with OPSCC (Saeed et al., 2022) or nasopharyngeal cancer (Zheng et al., 2022), researchers utilized the Transformer encoder to learn and combine multi-modality information, including

clinical and image data. These studies demonstrated that clinical data is essential for accurate outcome prediction. However, they did not compare the performance of their models with more simple methods for incorporating clinical data. Therefore, in this study, we compare three methods for incorporating clinical data into the TransRP model, as shown in Figure 1(b)(c)(d). The first two methods generally use the Transformer encoder to combine clinical and image features while the third utilizes a fully connected layer. We describe these methods in detail in next sections.

Method 1 (m1) was inspired by the study of Saeed et al. that used Transformer for PFS prediction in OPSCC patients (Saeed et al., 2022). As shown in Figure 1 (b), clinical predictors are first mapped to clinical features $f_{clc}$ with size $1 \times d$. $f_{clc}$ are then used as an additional embedded patch in the input to the Transformer encoder. The Transformer encoder is able to learn multi-modality information by learning the interaction between $f_{clc}$ and the other image embedded patches.

Method 2 (m2) was inspired by the study of Zheng et al. that used a combination of a CNN and Transformenr encoder for survival prediction in nasopharyngeal cancer patients (Zheng et al., 2022). As illustrated in Figure 1 (c), seven clinical predictors are first duplicated to size $n \times 7$, where $n$ is the original number of CNN-extracted feature patches. The duplicated clinical predictors are then concatenated with the image feature patches with size $n \times (1 \times 1 \times 1) \times C_I$ to produce combined features with size $n \times (1 \times 1 \times 1) \times (C_I + 7)$. The Transformer encoder then uses these combined feature patches for RFS prediction.

Method 3 (m3) in Figure 1 (d) uses a fully connected layer with 64 nodes to extract clinical features and then concatenates them to the second last layer of FCN in TranRP, which is a very simple way.

## 3. Experiments and Discussion

### 3.1. Data and preprocessing

The HECKTOR 2022 dataset (https://hecktor.grand-challenge.org/Data/) consists of 489 OPSCC patients who were collected with pretreatment PET/CT scans, GTV of primary tumors and lymph nodes, clinical data and clinical endpoint from seven different centers. The clinical predictors incorporated into TransRP include patient center (1: 7), age, gender (female: 0 vs. male: 1), weight, HPV status (positive: 0 vs. unknown: 1 vs. negative: 2), chemotherapy (no: 0 vs. yes: 1) and surgery (no or unknown: 0 vs. yes: 1). The clinical endpoint is recurrence-free survival (RFS) with events defined as local, regional recurrence and distant metastasis and time-to-event is defined in days starting with the end of radiation therapy. Patients are censored at the time of death or the time of the last follow up. Detailed data description can be found in (Andrearczyk et al., 2023). The data preprocessing and split are shown in Appendix B.

### 3.2. Implementation details

The $\lambda$ in Equation (3) was set to $5e^{-5}$. All models were trained using Pytorch 1.8.0 and MONAI 1.0.0 packages on a Tesla V100 GPU with 32G memory. A SGD optimizer with a momentum of 0.90 was initialized with a learning rate of 0.0002. The total training epoch was set to 200 and the learning rate decreased by a factor of 0.2 in 200th and 300th epoch,

respectively. Early stopping with a patience number of 25 epochs was used to select and save the model with the highest validation performance. The risk predictions of patients in the test set were the average ensemble of predictions from five-fold cross validation models. Data augmentation included random flipping in three directions with a probability of 0.5 (p = 0.5), random affine transformation (p = 0.5) and random elastic distortion (p = 0.2). Oversampling was implemented to keep patients with events and censored patients balanced in the training set.

### 3.3. Comparison of methods

We used TransRP(CNN-encoder name)-(the method of incorporating clinical data) to denote different TransRP models. Additionally, the performance of all TransRP models were compared with a DeepSurv (Katzman et al., 2018) model which uses only clinical data, a ResNet18-based model and a DenseNet121-based method (Wahid et al., 2021) which is the winner of the task 3 using CT/PET/GTVp in HECKTOR 2021 (Andrearczyk et al., 2021). To make a fair comparison, all models were trained using the same implementation details as shown in 3.2. ResNet18 and DenseNet121 were implemented using the default setting in MONAI 1.0.0 expect setting the stride in the first convolution layer to 2 in ResNet18. Clinical data was combined to ResNet18 and DenseNet121 using Method 3 (m3). The comparison of mean C-index ± standard deviation (std) in the validation sets, and the C-index and the average inference time for each patient in the test set are shown in Table 1.

Table 1: The C-indexes of different models and the average inference time in the test set.

| Model | Validation sets | Test set | Inference time (s) |
|---|---|---|---|
| DeepSurv (Katzman et al., 2018) | 0.614± 0.065 | 0.601 | 0.112 |
| ResNet18-m3 | 0.680±0.070 | 0.650 | 0.116 |
| DenseNet121-m3 (Wahid et al., 2021) | 0.679±0.071 | 0.676 | 0.140 |
| TransRP(ResNet18)-m1 | 0.658±0.047 | 0.624 | 0.127 |
| TransRP(ResNet18)-m2 | 0.695±0.081 | 0.644 | 0.125 |
| TransRP(ResNet18)-m3 | 0.674±0.069 | 0.686 | 0.123 |
| TransRP(DenseNet121)-m1 | 0.693±0.055 | 0.693 | 0.149 |
| TransRP(DenseNet121)-m2 | 0.694±0.068 | 0.671 | 0.150 |
| TransRP(DenseNet121)-m3 | 0.686±0.063 | 0.698 | 0.149 |

From the test C-index in Table 1, DenseNet121-m3 obtained a higher C-index of 0.676 than 0.650 of ResNet18-m3 and 0.601 of DeepSurv. When compared to ResNet18-m3, only TransRP(ResNet18)-m3 can obtain higher C-index of 0.686. Compared with DenseNet121-m3, TransRP(DenseNet121)-m1 and TransRP(DenseNet121)-m3 can obtain higher C-indexes of 0.693 and 0.698 (highest), respectively. In general, TransRP models which obtained both high mean and low std values of C-index in five-fold validation sets also achived high C-index in the test set such as TransRP(ResNet18)-m3, TransRP(DenseNet121)-m1 and TransRP(DenseNet121)-m3. The inference time of all models are within [0.11 to 0.15] seconds. Additionally, TransRP(CNN)-m3 models also showed better risk stratification ability than CNN models in the test set (Appendix C).

### 3.4. Ablation study

The ablation study was based on TransRP(DenseNet121)-m3 as it is the best predictive model, as shown in Table 1. The results presented in Table 2 indicate that excluding any component led to a decrease in the C-index.

Table 2: The ablation study of TansMP(DenseNet121)-m3.

| Clinical | CNN | Transformer | Model | Validation sets | Test set |
|---|---|---|---|---|---|
| ✓ | | | DeepSurv | 0.614±0.065 | 0.601 |
| | ✓ | ✓ | TransRP(DenseNet121) | 0.671±0.054 | 0.686 |
| ✓ | | ✓ | ViT-m3 | 0.640±0.072 | 0.656 |
| ✓ | ✓ | | DenseNet121-m3 | 0.679±0.071 | 0.676 |
| ✓ | ✓ | ✓ | TransRP(DenseNet121)-m3 | 0.686±0.063 | 0.698 |

## 4. Discussion and conclusion

From the test C-index comparison of DenseNet121-m3, ResNet-121-m3 and DeepSurv models (0.676 vs. 0.650 vs. 0.601) in Table 1, we can conclude that image features can improve RFS prediction in OPSCC patients and DenseNet121 is the better way of image feature extraction than ResNet18. When compared with CNN-based models including ResNet18-m3 and DenseNet121-m3, TransRP models combining clinical data by m1 and m2 (two methods using transformer to combine clinical and image features) only obtained one higher C-index of 0.693 in TransRP(DenseNet121)-m1. However, TransRP(ResNet18)-m3 and TransRP(DenseNet121)-m3 which use a fully connected layer to input clinical data obtained highest C-indexes 0.686 and 0.698, respectively. This finding contradicts previous studies which thought that transformer were superior in combining multi-modality information, such as clinical and imaging data, for the outcomes prediction in OPSCC or nasopharyngeal cancer patients (Saeed et al., 2022; Zheng et al., 2022). Best and worst predicted cases by TransRP(DenseNet121)-m3 are illustrated in Appendix D.

From ablation study in Table 2, excluding CNN-encoder (ViT-m3) caused a large C-index decrease from 0.698 to 0.656. This may be due to the Transformer's inability to extract detailed image features when the training set is small. It is also worth noting that TransRP(DenseNet121) which did not use clinical data, achieved a better C-index than DenseNet121-m3 (0.686 vs. 0.676). This suggests that, by learning both local detailed and global context features, TransRP without clinical data can even outperform CNN-based models that incorporate clinical data.

In conclusion, We propose a novel model named TransRP for the prediction of the recurrence-free survival in patients with oropharyngeal squamous cell carcinom. The TransRP uses a convolution neural network (CNN) to extract abundant CT and PET image features and then a Transformer to learn global contexts. TransRP obtained higher prediction metric than several clinical and CNN-based models. Additionally, we found that clinical data incorporated into TransRP using a simple fully connected layer showed better prediction than using transformer to combine clinical and image features.

## Acknowledgments

We thanks the data provided by MICCAI HECKTOR challenge and High Performance Computing cluster (Peregrine) provided by University of Groningen.

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

## Appendix A. The detailed architectures of ResNet18 and Transformer encoder

The detailed architectures of ResNet18 and Transformer encoder are shown in Figure 2.

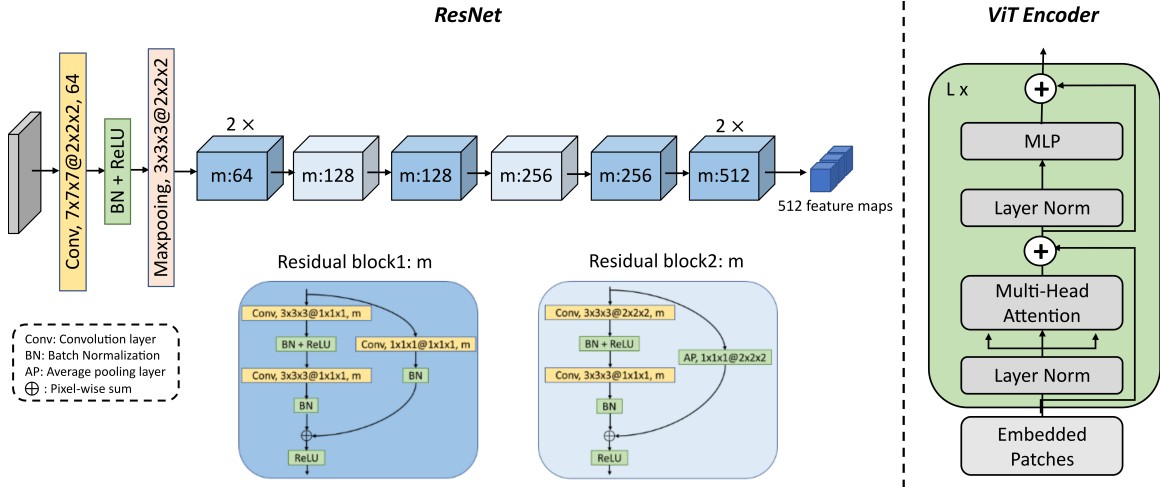

Figure 2: The detailed architecture of ResNet and ViT encoder. The notation "a×b×c@i×j×k" denotes a kernel size of a×b×c and a stride of i×j×k. m is the number of convolution kernels in residual blocks. MLP: Multi-layer peceptron.

## Appendix B. Data preprocessing

A bounding box containing GTV regions was generated with the size of $192 \times 192 \times 192 mm^3$ for each patient. The bounding box extraction method was inspired by the official code provide by HECKTOR challenge (https://github.com/voreille/hecktor). Firstly, the brain of each patient was segmented using PET threshing with the threshold 3-5. After getting brain segmentation, a point $p$ with position $(x, y, z)$ was determined as the point $30mm$ lower from the lowest voxel of the brain. Then, bounding box was obtained according to the position $(x - 96 : x + 96, y - 84 : y + 108, z - 72 : z + 120)$ where $x$, $y$ and $z$ denote coronal, sagittal and axial plans, respectively.

Then, the complete CT, PET and GTV image within the bounding box was extracted and resampled to a $2 \times 2 \times 2 mm^3$ resolution. The processed CT, PET and GTV were concatenated to obtain the input with size $96 \times 96 \times 96 \times 3$ for the model. Seven patients were excluded due to either abnormal PET values or no signal of brain in PET. The intensity values of CT and PET were truncated into ranges of [-200, 200] and [0, 25], respectively, and then were normalized to [0, 1]. Finally, 120 of 482 patients were randomly selected as the hold-out test set. Other 362 patients were split into five folds by stratified split of HPV status and RFS events number. The ratio of patients with events in the hold-out test set and the training set (362 patients) are 22.5% and 20.7%, respectively.

## Appendix C. Analysis of risk stratification by different models in the test set

The ability to stratify patients into a high (predicted risk $>$ the cut-off) group and a low risk group (predicted risk $<=$ the cut-off) was evaluated for different models in the test set. After obtaining the risk predictions of all 362 patients in the training set by average ensembling of five-fold models, the cut-off value was determined as the median value of these risks. The Kaplan-Meier curves (Kaplan and Meier, 1958) of two risk groups stratified by different models are shown in Figure 3. The log-rank tests (Mantel and Haenszel, 1959) were used to evaluate the RFS difference between high and low risk groups. Two tailed p-Value $< 0.05$ denotes the significant difference.

From Figure 3, there are significant differences ($p < 0.05$) between high and low groups stratified by ResNet18-m3, DenseNet121-m3, TransRP(ResNet18)-m3, TransRP(DenseNet121)-m1, TransRP(DenseNet121)-m2 and TransRP(DenseNet121)-m3. Compared with ResNet18-m3, TransRP(ResNet18)-m3 obtained smaller p-Value (0.003 vs. 0.019). Similarly compared with DenseNet121-m3, TransRP(DenseNet121)-m1 and TransRP(DenseNet121)-m3 achieved smaller p-Values of 0.022 and 0.001, respectively. The smallest p-Value of 0.001 was observed at TransRP(DenseNet121)-m3, which demonstrates that our TransRP model incorporating clinical data using a fully connected layer can stratify high and low risk groups with the largest difference.

## Appendix D. Best and worst predicted case illustration

To find the best and worst predicted cases, we ranked the patients in the test set according to their predicted risk scores by TransRP(DenseNet121)-m3. Then, the patient CHUP-013 was considered as the best predicted case because CHUP-013 was predicted with the highest risk score and he experienced a RFS event at a short time (175 days) after treatment. Additionally, CHUP-011 was selected as the worst predicted case because he has the second highest predicted risk score but he was censored at a long time (1749 days) after treatment. Their clinical and outcome data are displayed in Table 3 and imaging data are shown in Figure 4.

We did univariable Cox analysis using the clinical data in all 482 patients. Lower weight, alcohol use, higher PS, HPV negative status and no chemotherapy were found to be significant predictors of the shorter RFS. For the best predicted case CHUP-013, he has the above predictors: low weight, alcohol use and many primary tumor and lymoh nodes as shown in Figure 4. Thus, it is reasonable that our TransRP(DenseNet121)-m3 model predicted the highest risk score for CHUP-013. For the worst case CHUP-011, he also has alcohol use, HPV negative status and big primary tumors and lymph nodes as shown in Figure 4. Thus, it is also reasonable that this patient was predicted with the second highest risk score. However, this patient did not have event at least within 1749 days after treatment, which may be due to that he is a special case or other unknown possible predictors such as treatment plan were not considered in our model.

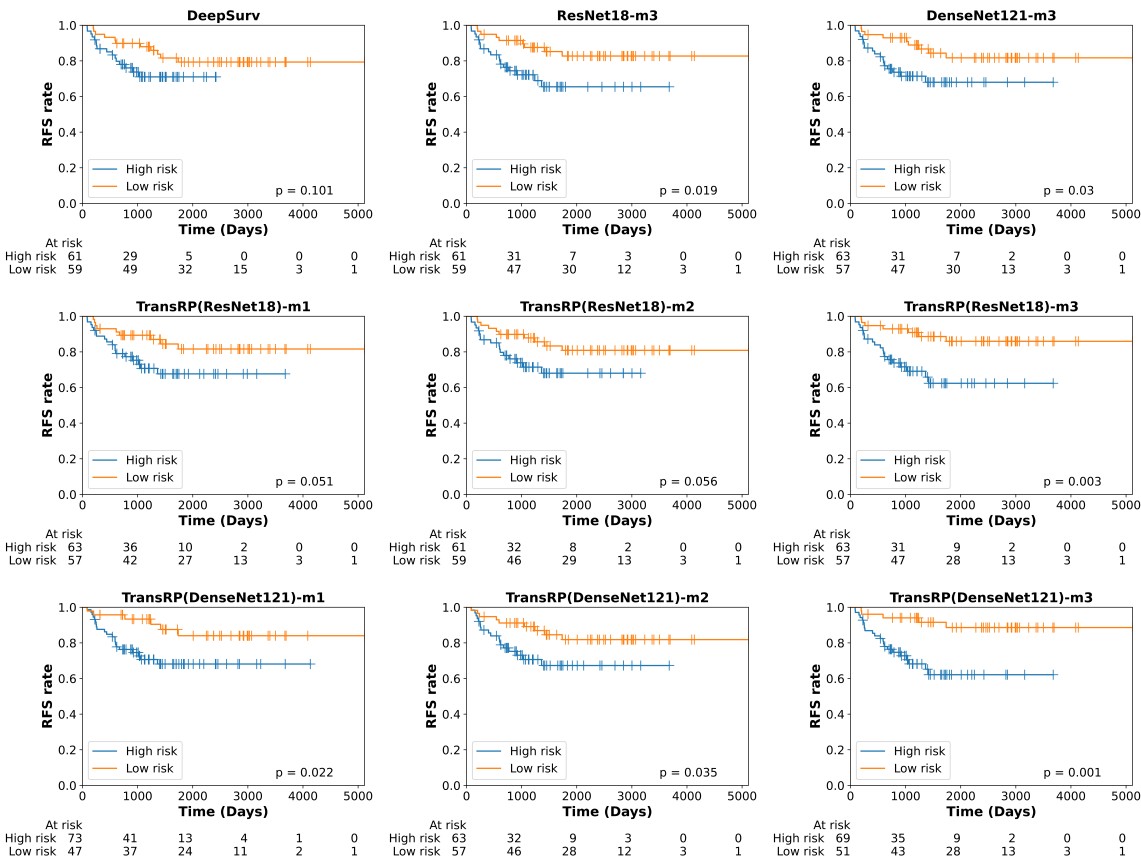

Figure 3: The Kaplan-Meier curves of high and low risk groups stratified by different models in the test set. All p-Values were obtained from the log-rank test

Table 3: The clinical data of patients CHUP-013 and CHUP-011. PS: performance status.

| ID Chemotherapy | Gender | Age | Weight | Tobacco | Alcohol | PS | HPV | Surgery |
|---|---|---|---|---|---|---|---|---|
| CHUP-013 yes | Male | 48 | 51kg | yes | yes | 1 | unknown | unknown |
| CHUP-011 yes | Male | 76 | 70kg | yes | yes | 0 | negative | unknown |

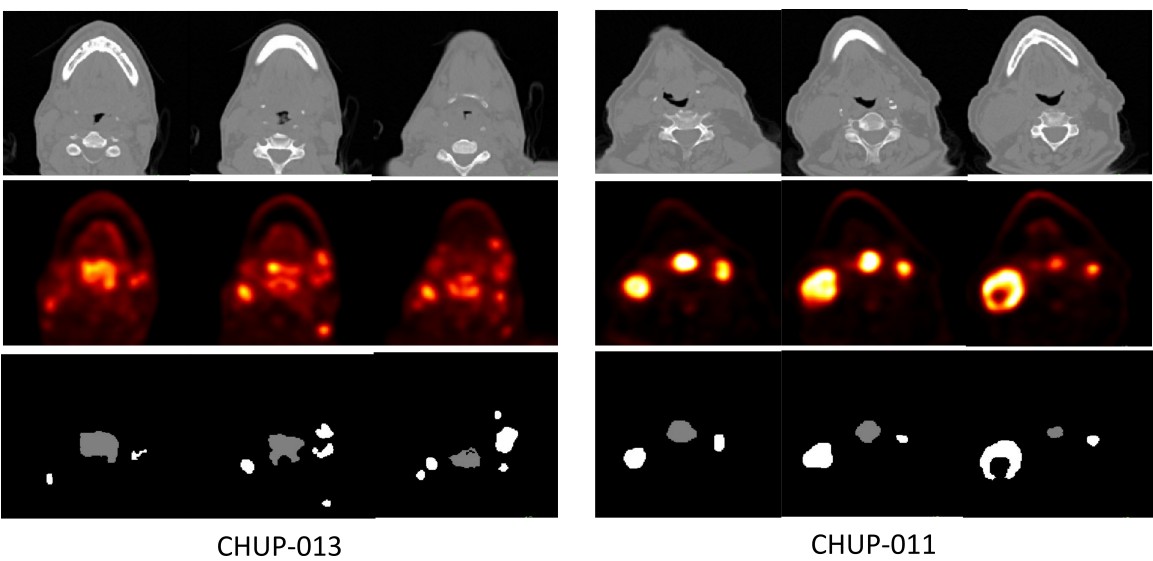

CHUP-013                    CHUP-011

Figure 4: The CT (1st row), PET (2nd row)and GTV (3rd row) images of CHUP-013 and CHUP-011.

