# OpenReview forum: "TransRP: Transformer-based PET/CT feature extraction incorporating clinical data for recurrence-free survival prediction in oropharyngeal cancer"
_MIDL.io/2023/Conference — MIDL 2023 Poster_

### Official Review · Reviewer_yDep · 2023-02-04

**Confidence:** 3
**Preliminary Rating:** 4
**Recommendation:** Poster

**Summary:**

The authors use a transformer-based method to fuse imaging data with clinical information to predict the recurrence-free survival time for oropharyngeal cancer patients. Their proposed method is benchmarked on the HECKTOR challenge dataset against DeepSurv, ResNet, and DenseNet. The performance is evaluated based on the C-index.

**Strengths:**

The clinical task and technical method are clearly motivated. The reader is conveniently guided through the text, and the figures contribute to the understanding of the method. Based on the experiments, the authors show a clear benefit of their approach regarding the C-Index over competing methods.

**Weaknesses:**

I would have appreciated an evaluation not solely based on a single metric (c-index). It would be interesting to see other metrics as well, and show some evaluation also demonstrating the variability across the five-fold CV.

**Deanonymize Review:**

no

**Detailed Comments:**

I appreciate the very detailed experiment description and the training details. The ablation study further contributes to understanding the influence of the different components.

Typo:
- Last sentence in 3.1.: [...] were stratified into five folds [...]

**Paper Type:**

methodological development

**Questions To Address In The Rebuttal:**

- Please consider adding other evaluation metrics also showing the variability across folds. This could be e.g. error measures with standard deviations, Bland-Altman plots etc.
- Could your approach also be directly used in a regression task, i.e. directly regressing the RFS?

---

### Official Review · Reviewer_p2Tk · 2023-02-05

**Confidence:** 5
**Preliminary Rating:** 4
**Recommendation:** Poster

**Summary:**

This paper presents a transformer based model for predicting recurrence free survival in OPSCC patients. The method is called TransRP, consisting of a CNN based encoder to extract PET/CT features, and a Transformer encoder to learn global context, and fully connected network to incorporate non-imaging features for RFS prediction. More than 2% improvement was observed in concordance index.

**Strengths:**

-- clinically important problem is addressed
-- a new network is proposed, based on combined CNN and Transformers.
-- clinical data is integrated into the decision mechanisms in addition to imaging data
-- three comparisons were made with some sound approach


**Weaknesses:**

-- figure 1 is misleading, figure 2 is more representative of what is being done, it I s connection should be more clearly explained, or even combined figure 1 and figure 2.
-- all the methods presented are available in the literature, there is no novelty here but application is interesting and incrementally novel,
-- combination of CNN and Transformer is shown to be superior to other methods (Table 2), why is so? why not purely transformer methods?
-- what are the best and worst case illustration?
-- what clinical parameters are integrated, and good for predicting patient prognosis ?


**Deanonymize Review:**

no

**Detailed Comments:**

weaknesses are self-explanatory comments, indicating both detailed comments and questions.



**Paper Type:**

validation/application paper

**Questions To Address In The Rebuttal:**

-- figure 1 is misleading, figure 2 is more representative of what is being done, it I s connection should be more clearly explained, or even combined figure 1 and figure 2.
-- all the methods presented are available in the literature, there is no novelty here but application is interesting and incrementally novel,
-- combination of CNN and Transformer is shown to be superior to other methods (Table 2), why is so? why not purely transformer methods?
-- what are the best and worst case illustration?
-- what clinical parameters are integrated, and good for predicting patient prognosis ?

---

### Official Review · Reviewer_1Cgg · 2023-02-06

**Confidence:** 4
**Preliminary Rating:** 3

**Summary:**

In this paper, the authors proposed a Transformer-based model for predicting recurrence-free survival (RFS) in OPSCC patients, called TransRP, which consists of a CNN encoder to extract rich PET/CT image features, a Transformer encoder to learn global context features, and a fully connected network to incorporate clinical data for RFS prediction. In general, it is well-written, and the experimental results are encouraging.

**Strengths:**

1. A novel Transfomer-based model named TransRP is proposed for prediction of RFS in OPSCC patients based on PET/CT/GTV and clinical data.
2. TransRP combines a CNN-encoder for the extraction of rich image features and a Transformer-encoder for learning global contexts.
3. In general, it is well-written, and the experimental results are encouraging.

**Weaknesses:**

1. The TransRP architecture shown in Figure 1 is not innovative, it is just a combination of CNN and transformer. And, what is the “TransMP” in the figure caption of Figure 1? There is no explanation given in the manuscript.

2. The authors claim to have performed a five-fold cross-validation, therefore, the standard deviation between the five results should be given in the experimental results to further assess the stability of the proposed methods.

3. We all know that the execution efficiency of the transformer is significantly lower than that of the CNN, and the authors should give the execution efficiency to evaluate the different methods more comprehensively.

**Deanonymize Review:**

no

**Paper Type:**

both

**Questions To Address In The Rebuttal:**

1. Carefully revise the manuscript to avoid spelling errors and improve writing details

2. The standard deviation between the five cross-validation results should be given in the experimental results to further assess the stability of the proposed methods.

3. Give the execution efficiency to evaluate the different methods more comprehensively.

---

### Official Review · Reviewer_QUqW · 2023-02-06

**Confidence:** 4
**Preliminary Rating:** 1

**Summary:**

This study proposes a Transformer-based model called TransRP for predicting recurrence-free survival in oropharyngeal squamous cell carcinoma (OPSCC) patients. TransRP uses a combination of a CNN encoder to extract image features, a Transformer encoder to learn global context features, and a fully connected network to incorporate clinical data. The study was conducted using a public dataset and the results showed that TransRP achieved the highest concordance index of 0.698 compared to other clinical and CNN-based methods. The study found that incorporating clinical features with image features performed better than using the Transformer encoder to extract features from both clinical and image features.

**Strengths:**

- Recurrence-free survival prediction in oropharyngeal cancer is an important and relevant clinical question.
- Positron emission tomography–computed tomography are used to train the model.
- Integration of clinical data to the image-based model.

**Weaknesses:**

- The authors claim that they propose a novel transformer-based model named TransRP, yet, it is very hard to find what is novel compared to other work using transformer (Saeed et al., 2022, Zheng et al., 2022 ...etc). The authors state that in previous works "they did not compare the performance of their models with more simple methods for incorporating clinical data". However, this does not add novelty to the transformer-based model as claimed by the authors.
- The paper is badly written and needs substantial improvement. Please describe clearly what is the main contributions of the paper compared to what was done previously. Please do not conclude or discuss in the results section, and only report results. For instance "In conclusion, Method 3 (m3) which uses a fully connected layer to input clinical data into TransRP obtained highest C-indexes whether the CNN-encoder is ResNet18 or DenseNet121." should be in conclusion not in Results.
- Please focus the paper on the main contribution, what is novel in this work? what was challenging? how did you solve it? The current work only uses previous transformer-based model and add some experimental analysis.
- The authors used 5 cross validation and then report the results on a randomly selected hold-out dataset of 128. What was the ratio of event/no event in the hold-out dataset? How did you choose the best model? Did you use ensembling? What are the results on the cross-validation? are the models robust to changes ? Please show the mean and standard deviation.


**Deanonymize Review:**

no

**Paper Type:**

methodological development

**Questions To Address In The Rebuttal:**

- Please see above in Weaknesses section.
- In figure1, do you mean TransRP instead of TransMP ?
- "Finally, 120 of 482 patients were randomly selected as the test set and 362 patients were stratified into five-fold in terms of HPV status and RFS events number."
- The writing of the manuscript needs improvement. For example, this is not a sentence "Because TransRP has two CNN-encoder options (ResNet18 and DenseNet121) and three optional methods (m1, m2 and m3) of incorporating clinical data."
- "From Table 1, it is obvious that DenseNet121-m3 performed better than ResNet18-m3". Please use a scientific language. Also, what do you mean by obvious? Is the difference between 0.676 and 650 significant ?
- The 3D input volume is relatively big. Can you comment on the training/inference time? Compared to other models?

---

### Meta-Review · Area_Chair_NNhB · 2023-02-24

**Recommendation:** Accept (Poster)
**Confidence:** 4

**Metareview:**

The authors have made some progress in addressing the concerns raised by the reviewers, specifically regarding the motivation and experimental performance. However, one concern remains regarding the technical novelty of the transformer, and a thorough proofreading is necessary to ensure the quality of the manuscript. Overall, the study presents a novel clinical application. While it meets the minimum requirements for publication, addressing the remaining concerns would further enhance its impact and credibility.

---

### Meta-Review · Program_Chairs · 2023-03-01

**Recommendation:** Accept (Poster)
**Confidence:** 5

**Metareview:**

The PC felt that this was a novel clinical application, so despite some concerns for the novelty of the transformer, the decision was to accept this paper as a Poster.